# Characteristics of COVID-19 Inpatients in Rehabilitation Units during the First Pandemic Wave: A Cohort Study from a Large Hospital in Champagne Region

**DOI:** 10.3390/biology11060937

**Published:** 2022-06-20

**Authors:** Amandine Rapin, Peter-Joe Noujaim, Redha Taiar, Sandy Carazo-Mendez, Gaetan Deslee, Damien Jolly, François Constant Boyer

**Affiliations:** 1Faculté de Médecine, Université de Reims Champagne Ardennes, UR 3797 VieFra, 51097 Reims, France; arapin@chu-reims.fr (A.R.); djolly@chu-reims.fr (D.J.); fboyer@chu-reims.fr (F.C.B.); 2Service de Médecine Physique et de Réadaptation, Hôpital Sébastopol, CHU de Reims, 51092 Reims, France; scarazo-mendez@chu-reims.fr; 3Unité D’aide Méthodologique, Pôle Recherche et Santé Publique, Hôpital Robert Debré, CHU de Reims, 51092 Reims, France; pjnoujaim@chu-reims.fr; 4Matériaux et Ingénierie Mécanique MATIM, Université de Reims Champagne Ardennes, CEDEX 2, 51687 Reims, France; 5Service des Maladies Respiratoires, Hôpital Maison Blanche, CHU de Reims, 51092 Reims, France; gdeslee@chu-reims.fr; 6Inserm UMR-S1250, P3Cell, University of Reims Champagne-Ardenne, SFR CAP-SANTE, 51092 Reims, France

**Keywords:** COVID-19, rehabilitation centers, physical and rehabilitation medicine, healthcare trajectory, functioning, frailty

## Abstract

**Simple Summary:**

Analyzing how healthcare institutions adapted to the first wave of the COVID-19 pandemic will make it possible to better prepare for future crises. Analyzing the characteristics of patients admitted to rehabilitation units during the first wave makes it possible to better understand rehabilitation needs during a pandemic crisis. The characteristics of all patients admitted to a single, large university hospital in Northeast France for SARS-CoV-2 infection during this period are described. The initial severity of infection and advanced age were associated with referral to inpatient rehabilitation units. The proportion of patients who had access to inpatient rehabilitation units was lower than expected in this study, which raises questions about the deployment of rehabilitation services. The availability and organization of rehabilitation plans in acute care units and post-acute care are key elements to anticipate in the event of a pandemic crisis.

**Abstract:**

**Background**: Data describing patients hospitalized in medical rehabilitation wards after the acute phase of COVID-19 could help to better understand the rehabilitation needs in the current pandemic situation. **Methods**: Cohort including all patients with COVID-19 hospitalized in a single, large university hospital in Northeast France from 25 February to 30 April 2020. **Results**: 479 patients were admitted with COVID-19 during the study period, of whom 128 died (26.7%). Among the 351 survivors, 111 were referred to rehabilitation units, including 63 (17.9%) referred to physical and rehabilitation medicine (PRM) units. The median age of patients referred to rehabilitation units was 72 years. Patients who had been in intensive care, or who had had a long hospital stay, required referral to PRM units. Two biomarkers were associated with referral to rehabilitation units, namely, elevated troponin (*p* = 0.03) and impaired renal function (*p* = 0.03). Age was associated with referral to PRM units (*p* = 0.001). **Conclusions**: Almost one-third of COVID-19 patients required post-acute care, but only one-fifth had access to PRM units. The optimal strategy for post-acute management of COVID-19 patients remains to be determined. The need for rehabilitation wards during a pandemic is a primary concern in enabling the long-term functioning of infected patients.

## 1. Introduction

COVID-19 is a respiratory infection caused by the SARS-CoV-2 virus, and the main symptoms are fever and cough. About 15% of patients with symptomatic infection will have a severe form of infection and around 5% require intensive care [1]. Several factors have been shown to be related to mortality, such as obesity, arterial hypertension, diabetes, and cardiovascular disease [2,3]. Biological markers such as troponin or renal function are also related to the severity of infection [4,5].

During the first wave of the COVID-19 pandemic, there was a huge influx of patients to hospitals, with widely varying levels of severity. The rapid increase in incidence required correspondingly high numbers of intensive care unit (ICU) beds, which were soon in short supply [6,7]. Hospitals worldwide had to adapt their organization and resources to facilitate management pathways and provide acute care for a maximum number of COVID-19-infected patients. Different countries implemented different adaptation strategies to strengthen their capacities to respond to the epidemic, for example, by mobilizing medical students, or through cooperation with primary care [8,9]. At the time, knowledge of the virus was scanty, and information was lacking about the different forms and severity, resulting in uncertainty surrounding optimal management, despite the early publication of recommendations [10].

The organization of the public hospital system in France provides a first-line reserve of acute hospitalization wards in medicine, surgery, and obstetrics, including medical and surgical ICUs. Second-line resources include polyvalent rehabilitation units (PRU) or specialized rehabilitation units (e.g., physical and rehabilitation medicine (PRM) units).

During the first pandemic wave, the northeast of France was one of the regions hit hardest, with high incidence rates [11]. Initially, the urgency lay in ensuring survival for the highest number of patients, relegating questions about short-, medium-, and long-term functioning to the background. Experiences from prior epidemics caused by infectious agents such as SARS-CoV-1 and MERS-CoV had already suggested that there would be consequences on overall health in the medium to long term, including physical, mental, cognitive, and functional repercussions, notably for the most severely affected patients [12,13,14]. The literature about the current COVID-19 disease confirms that the same consequences are to be expected [15,16]. Therefore, PRM specialists expected to receive many patients within a short time, all requiring personalized and multicomponent rehabilitation due to the wide heterogeneity in symptoms.

The need for post-acute rehabilitation, as well as the need to free up beds in medical wards for new patients, prompted widespread recourse to PRU or PRM units. Targeted early rehabilitation in acute wards or the ICU was also widely implemented. PRM units adapted the organization of their care delivery to cater for the sudden and intense demand for rehabilitation services for COVID-19 patients [17].

In the context of emergency re-organization, and without knowing the outcome in the post-acute phase, the medical community did its utmost to treat patients as well as possible. Patients who were referred to rehabilitation units were the patients who were deemed to have the highest priority. The study of these criteria would make it possible to understand the choices made, analyze the relevant factors driving those choices, as well as highlight areas where there may be room for improvement. Previous studies have investigated the characteristics of patients admitted to the ICU who were discharged and readmitted to an acute care unit [18,19,20]. Wiertz et al. described the characteristics of patients admitted to rehabilitation units, but without comparing them to patients not referred to rehabilitation units [21].

The aim of this study is to describe the profiles of COVID-19 patients referred to rehabilitation units during the first epidemic wave in the University Hospital of Reims, France, to better understand rehabilitation needs during a pandemic crisis.

## 2. Materials and Methods

### 2.1. Study Design

The REIMS-COVID19 cohort is a follow-up study of all patients admitted to the University Hospital of Reims, Northeastern France, during the first pandemic wave (from 25 February to 30 April 2020). Due to the fact that the pandemic first began in France in the northeastern and north-central departments, with temporal progression from these departments towards the rest of country [22], the period selected corresponds to the most intense period of the first wave, as there was a marked drop in the number of patients being admitted after the 30th April, as the effects of the national lockdown began to become apparent.

Each day, the cases were identified prospectively from the hospital medical records database, which was authorized by the French “Commission Nationale Informatique et Libertés” (CNIL) under number 1 118 523. The data were recorded and then updated each day in a specific informatics database.

The results are reported in compliance with the Strengthening the Reporting of Observational studies in Epidemiology (STROBE) statement for cohort studies [23].

### 2.2. Study Population

Patients with a confirmed diagnosis of COVID-19 were identified in the hospital informatics database based on the information from their medical files and biological results.

Inclusion criteria were: (1) hospitalization for COVID-19 infection during the first wave of the pandemic, i.e., from 25 February to 30 April 2020; (2) aged 18 years or over; (3) affiliation to or beneficiary of a health insurance regime; (4) COVID-19 diagnosis confirmed by a positive PCR test and/or compatible imaging; (5) non-opposition to the use of medical data for research purposes. There were no specific exclusion criteria.

This study was approved by the Ethics Committee (CPP Ile de France III) under the number CPP 3838-RM. The study was registered with ClinicalTrials.gov under the identifier NCT04553575. According to the protocol, patients were followed-up at 6, 12 and 24 months after diagnosis, with a medical consultation at each follow-up timepoint.

### 2.3. Study Measurements

Data were retrospectively collected from the patients’ computerized medical records. The biological data in the medical files were all validated by the physicians responsible for the laboratory at the time of analysis, before being entered into the medical file. Anthropometric data, such as weight and height, were re-read and verified by the caregivers working with the patients. Age, sex, anthropometric variables, medical history, comorbidities (with calculation of Charlson’s comorbidity index [24]), biological results during the acute phase, initial clinical presentation including evaluation of severity, Early Warning Score (EWS) [25], need for oxygen therapy, breathing rate, occurrence of pulmonary embolism (PE) during the acute phase, as well as information about treatment (admission to ICU), antibiotic therapy, corticosteroid therapy, anticoagulant therapy, length of stay, and prescription of acute rehabilitation, were recorded.

Hematological indicators (leukocytes and lymphocytes) were estimated using the Sysmex^®^XN1000 analyzer (Sysmex, Roissy CDG, France). All biochemical indicators were estimated using the Roche analyzer with Roche reagents (Roche Diagnostics, Meylan, France). Creatinine was estimated with an enzymatic assay protocol, and glomerular filtration rate (GFR) was calculated using the CKD-EPI equation. C-reactive protein (CRP) and albuminemia were estimated by immunoturbidimetry. Troponin was estimated by electrochemiluminescence.

Patients were considered to be at risk of severe forms of COVID-19 if they were aged >70 years, had immunosuppression, arterial hypertension, diabetes, body mass index (BMI) ≥ 40 kg/m², a history of cardiovascular disease, cirrhosis Child–Pugh class B or C, chronic respiratory disease, renal failure requiring renal replacement therapy, cancer, and/or were pregnant.

In accordance with the recommendations of the French Society of Anesthesia & Intensive Care Medicine (SFAR) [26], initial clinical presentation was considered to be severe in the case of fever and SpO2 < 90% in room air, breathing rate > 30/min, need for invasive or non-invasive ventilation, or in the case of respiratory or cardio-circulatory organ failure.

Whenever possible, the patient’s functional capacity prior to infection was calculated according to Katz’s activities of daily living (ADL) and patients were classified as completely autonomous (score = 6) or not (score < 6) [27]. The implementation of early rehabilitation during the acute stay was also recorded.

### 2.4. Statistical Analysis

No imputation was performed for missing data. The data distribution was studied using the Shapiro–Wilk test. In the case of non-normal distribution, quantitative data were described using medians and first and third quartiles (25th and 75th percentiles), and qualitative data were described using numbers (percentages).

Patients admitted to PRM units were compared to patients discharged directly to their homes after the acute phase, and to patients admitted to PRU. Qualitative variables were compared with the chi-square test, and quantitative variables using the Student t or Mann–Whitney U test, according to the normality of distribution. A *p*-value < 0.05 was considered statistically significant. All analyses were conducted using R studio^®^ Version 4.0.5 (RStudio, Boston, MA, USA).

## 3. Results

### 3.1. Population Distribution

A total of 499 patients were admitted during the study period, of whom 479 had confirmed SARS-CoV-2 infection. Among these, 128 (26.7%) died in hospital. Among the 351 survivors, 223 (63.5%) were discharged to their homes, 111 (31.6%) were referred to post-acute units (63 (17.9%) to PRM units and 48 (13.7%) to PRU), 6 (1.7%) were transferred to a facility closer to their home, and the discharge destination was missing for 1 patient. The flowchart of the study population is shown in Figure 1.

### 3.2. Population Characteristics

The characteristics of the study population according to their discharge destination are described in Table 1. Among 65 patients who were hospitalized in the ICU during the acute phase, 44.6% were discharged directly to their homes without inpatient rehabilitation. Among those who were referred to PRM units, 42.9% had been in the ICU during the acute phase. Patients addressed to PRM units were older and more at risk of severe forms of disease than patients discharged to their homes. Patients addressed to PRU were older and had a higher BMI than patients referred to PRM units.

The median length of stay in acute care was 15 days [10–32.5] for patients who were referred to PRM units, versus 8 days for those who were discharged directly to their homes.

Data regarding the clinical presentation, biology results, and treatment during the acute phase are described in Table 2.

Patients with acute renal failure were more frequently referred to PRM units and had higher levels of troponin at the acute phase than patients discharged to their homes, but had lower CRP and troponin levels than patients referred to PRU.

Data regarding functional capacity prior to COVID-19 infection were available for 259 patients (74%). An ADL score < 6 was found in 10 patients (15.9%) who were referred to PRM units, 19 (39.6%) in PRU, and 31 (13.3%) patients discharged directly to their homes.

Among the overall population of 479 patients, 80 had early rehabilitation during their acute hospital stays, namely, 9 patients (7%) who died, 26 patients (41.3%) who were referred to PRM units, 8 patients (16.7%) who were discharged to PRU, 35 patients (15%) who were discharged to their homes, and 2 patients who were transferred to care facilities closer to their homes.

A comparison of patients discharged to their homes and those referred to PRU + PRM units is shown in the Appendix A.

## 4. Discussion

This study investigated the healthcare trajectories of patients hospitalized for COVID-19 during the first wave in France and provides insights into post-acute rehabilitation needs in a geographical region that was hit particularly hard in the early phase of the pandemic [11].

In this cohort, one-third of patients who survived the acute phase subsequently moved on to a post-acute unit, whereas 60% were discharged directly to their homes. Almost one-fifth had access to PRM units. Several studies have sought to quantify rehabilitation needs after COVID-19. Halpin et al. reported that 60.2% of patients had fatigue and 42.6% had dyspnea between 4 and 8 weeks after discharge [28]. Nakayama and al. reported that 62.6% of patients had persistent symptoms after discharge [29]. Regarding participation, Fugazzaro et al. found that 76% of patients had participation restrictions, as assessed by the Reintegration to Normal Living Index, at 3 months after the acute infection [30]. In light of these proportions, it seems insufficient that only one-fifth of patients are receiving PRM. This shortfall was also highlighted by Daunter et al., who reported that 45.2% of patients had functional decline impacting their discharge after hospitalization, and 40.6% were never assessed by a PRM physician, physical therapist, occupational therapist, or speech therapist during their hospital stay [31].

In the present study, the patients who were referred to PRM units were those who had had the most severe forms of disease, required ICU admission, and had a longer median length of acute stay. The association between ICU and PRM unit admission is consistent with rehabilitation needs after ICU [32]. The longer median length of stay results in prolonged immobility with potential consequences, such as sarcopenia [33], creating further rehabilitation needs.

Older patients and patients at risk of severe forms of disease due to cumulative risk factors were more often referred to PRM units in the present study, whereas patients with a single risk factor for disease severity, such as obesity or diabetes, were less frequently referred. A potential explanation can be found in the frailty definition, related to age and comorbidities. Hägg et al. [34] showed an association between the level of frailty, as measured by the Clinical Frailty Scale, and the discharge after hospitalization for COVID-19 disease. Comorbidities were not associated with discharges to patients’ homes in that study, which is consistent with our results.

Certain biomarkers known to translate the severity of COVID-19 infection, such as troponin or renal function [4,5], were also associated with transfer to PRM units. Different mechanisms may explain the increase in troponin level at the acute phase of COVID-19, including non-ischemic myocardial disease processed due to the severity of infection [35]. Acute renal failure is also related to severity and age, which is consistent with the higher age in PRM patients [36]. Thus, severity may not be the only explanatory factor.

Indeed, it should be noted that almost 45% of patients who had been in the ICU during their acute stay were discharged to their homes without specialist advice from a rehabilitation unit. Furthermore, the initial severity of disease did not distinguish between those requiring PRM units or those discharged to their homes. This, again, suggests that the severity of the infection alone cannot explain the referral (or lack of referral) to rehabilitation units after the acute phase in the first wave.

Several hypotheses can be proposed to explain these findings. First, the patient’s healthcare trajectory is not solely dependent on the patient’s characteristics [37]. The availability of care opportunities in the geographical region under study also plays a major role. The rapid and massive influx of patients necessitated thorough upheaval of healthcare delivery in the early stages of the pandemic. Therefore, PRM units admitted patients who had lost autonomy, or who were unable to return home immediately because of a risk of contagion, in order to avoid saturation of hospital beds. This contributed to the saturation of PRM units. The problem of the availability of places during the first wave has also been studied in other types of units, such as the ICU, with a need for rationing [6]. This could explain why almost half of patients who had been in the ICU were discharged to their homes. Home return was acceptable, assuming that patients would receive rehabilitation in a home setting from community professionals [38]. Ahmad et al. [39] described a post-COVID-19 care center that was established to identify patients with disease sequelae with the goal of delivering early multidisciplinary rehabilitation services. Since not all rehabilitation needs can be covered, systematic evaluation or open consultations should be developed within an overall strategy to offer early rehabilitation interventions [40] and to maintain the continuum of care [41].

Another hypothesis purports that there may be no absolute relation between the severity of the initial viral infection and the expected functional consequences. Several studies have failed to find an association between the initial severity of SARS-CoV-2 infection and the functional consequences in the medium term (up to 3 months) [42,43,44,45]. The need for PRM stems from a need for complex rehabilitation in hopes of achieving functional recovery and autonomy compatible with a patient’s return to their home. The criteria describing the severity of initial infection, or biomarkers reflecting severity, mainly characterize the initial phase and the risk of mortality [46] more than the risk of transient or permanent functional consequences (e.g., prolonged or persistent post-COVID-19 syndrome). Finally, admission to PRM units also depends on factors such as the organization of rehabilitation services, the patient’s lifestyle or living conditions, as well as the opinion of the treating physician and/or the physician who approves the rehabilitation admission. These are all factors that can be difficult to measure and/or regulate [47].

In the present study, patients admitted to PRU were older, compared to those addressed to PRM units, while those discharged to their homes were the youngest. The severity of COVID-19 is known to be associated with age [48]. The functional impact of an acute health event or stress is more pronounced with increasing age, taking into account frailty or pre-infection functional reserve [49]. Patients with less functional reserve have a more limited capacity to participate in intense rehabilitation programs. Some of these frailer patients were preferentially oriented towards PRU, where the rehabilitation activities are less intense and are spread out over longer intervals than in PRM units.

This study has some limitations. Firstly, the data were collected from the hospital informatics database and, therefore, precluded the collection of standardized information about the pre-admission capacity or functional reserve of the patients. Furthermore, the high rate of missing data for some variables should prompt caution in the interpretation of the findings. The data in this study were collected during the first wave of the pandemic, when certain markers now known to be related to severity (such as D-dimers) were not systematically measured. Finally, we did not perform multivariate analysis, as it did not appear legitimate to construct a predictive model for a local situation in a single institution with its own specific organization.

## 5. Conclusions

In summary, a clear description of the need for in-hospital rehabilitation and the criteria for admission to rehabilitation units are essential to help improve the organization of healthcare delivery beyond the acute phase. Although the severity of infection is an important criterion, it cannot fully explain the choice to orient a given patient to rehabilitation. Older age was associated with greater rehabilitation needs in the first wave of the COVID-19 pandemic, but comorbid conditions such as hypertension, diabetes, or obesity do not appear to be related to admission to rehabilitation units. Other factors clearly need to be considered to enhance the understanding of post-acute healthcare pathways.

Organizational conditions for healthcare delivery and availability of services/beds, or the patient’s functional capacity, must be concerns. Pandemic strategies for adapting access to rehabilitation should be developed, such as early consultation and multidisciplinary evaluation of discharged patients, to maintain the continuum of care.

The absence of a clear relationship between the severity of disease and functional impact should also be taken into account, and the predictors of functional alteration remain to be identified. In times of healthcare crisis or emergency, the systematic evaluation of functional status at admission to acute care would constitute a significant step towards improving practices, and would help with subsequent assessment of the need for rehabilitation.

These are important avenues for further reflection and research, to define and plan the healthcare trajectory of patients after acute hospitalization, and to tailor rehabilitation services appropriately.

## Figures and Tables

**Figure 1 biology-11-00937-f001:**
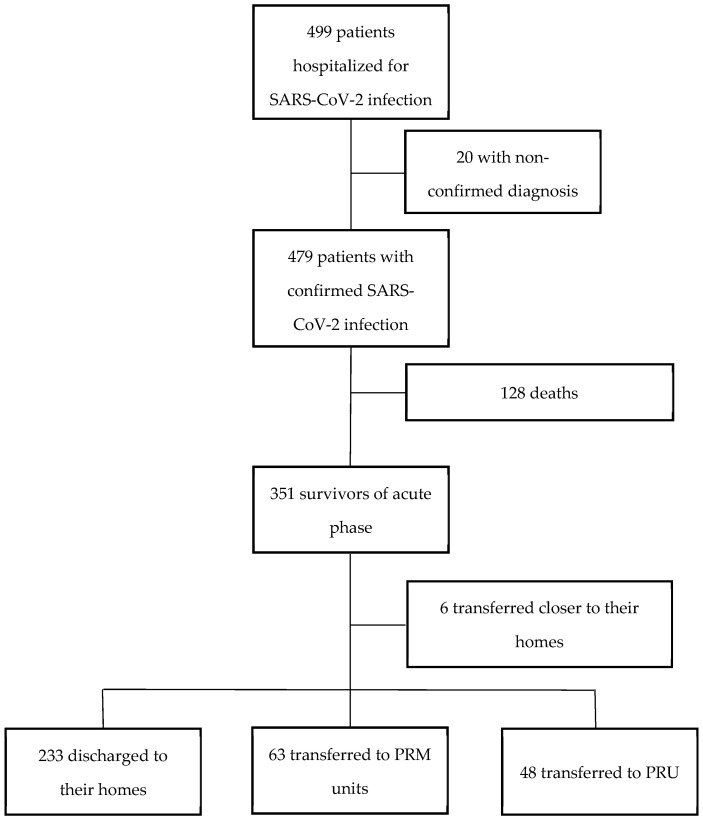
Flowchart of the study population.

**Table 1 biology-11-00937-t001:** Characteristics of patients according to discharge destination (PRM, PRU, home).

Characteristic	Home (A) *n = 233*		PRM Units (B) *n = 63*		*p Value* *(A) vs. (B)*	PRU (C) *n = 48*		*p Value* *(B) vs. (C)*
Women *(n, %)*	120	(51.5%)	28	(44.4%)	*0.320*	22	(45.8%)	*0.884*
Age, years *(median, [Q1–Q3])*	67	[53–79]	72	[68–81]	*0.001*	82	[72.8–85]	*0.002*
BMI Kg/m^2 ‡^ *(median, [Q1–Q3])*	29	[25–32]	27	[23–32.2]	*0.293*	24	[22–28]	*0.010*
BMI ≥ 30 Kg/m^2 ‡^ *(n, %)*	13	(5.6%)	4	(6.3%)	*0.999*	0	(0.0%)	*0.129*
Current smokers * (*n*, %)	14	(6.0%)	3	(4.8%)	*0.572*	1	(2.1%)	*0.999*
Arterial hypertension (*n*, %)	117	(50.2%)	38	(60.3%)	*0.572*	29	(60.4%)	*0.885*
Diabetes (*n*, %)	54	(23.2%)	21	(33.3%)	*0.253*	11	(22.9%)	*0.197*
Chronic respiratory disease (*n*, %)	40	(17.2%)	14	(22.2%)	*0.610*	10	(20.8%)	*0.810*
Renal failure requiring RRT (*n*, %)	14	(6.0%)	3	(4.8%)	*0.769*	4	(8.7%)	*0.697*
Charlson score *(median, [Q1–Q3])*	1	[0–2]	1	[0–3]	*0.586*	2	[1–4]	*0.093*
At risk of severe form of disease (*n*, %)	194	(83.3%)	59	(93.65%)	*0.038*	46	(95.8%)	*0.697*

PRM = physical and rehabilitation medicine; PRU = polyvalent rehabilitation units; BMI = body mass index; Q = quartile (Q1 = 25th percentile and Q3 = 75th percentile); RRT = renal replacement therapy. ^‡^ 82 missing data, * 40 missing data.

**Table 2 biology-11-00937-t002:** Clinical presentation, biology results, and treatment of patients according to discharge destination.

	Home (A) *n = 233*		PRM Units (B) *n = 63*		*p Value* *(A) vs. (B)*	PRU (C) *n = 48*		*p Value* *(B) vs. (C)*
* **Clinical presentation** *								
Severe clinical presentation *(n, %)*	60	(25.8%)	22	(34.9%)	*0.149*	13	(27.1%)	*0.379*
Early warning score ^†^ *(median, [Q1–Q3])*	6	[3–8]	7	[4–9]	*0.091*	8	[5–10]	*0.343*
Oxygen therapy at admission *(n, %)*	89	(38.2%)	30	(47.6%)	*0.060*	23	(47.9%)	*0.782*
Breathing rate *(median, [Q1–Q3])*	22	[18–26]	20	[17.8–28]	*0.779*	20	[18–25.8]	*0.875*
Pulmonary embolism *(n, %)*	10	(4.3%)	4	(6.3%)	*0.509*	3	(6.2%)	*0.999*
* **Biology during acute infection** *								
Leukocytes, G/L *(median, [Q1–Q3])*	6.9	[4.9–9]	6.55	[4.8–8.2]	*0.250*	6	[3.8–9.8]	*0.999*
Lymphocytes < 1.5 G/L *(n, %)*	179	(76.8%)	53	(84.1%)	*0.09*	35	(72.9%)	*0.22*
GFR < 60 mL/min/kg *(n, %)*	56	(24%)	24	(38.1%)	*0.03*	18	(37.5%)	*0.98*
Albuminemia, g/L ^₮^ *(median, [Q1–Q3])*	35	[31–38]	33	[30.8–36]	*0.072*	30	[26–37]	*0.121*
CRP mg/L *(median, [Q1–Q3])*	61.6	[22.5–120]	90,3	[30.4–157]	*0.096*	37.7	[14.2–91.4]	*0.032*
Troponin, ng/L ^¥^ *(median, [Q1–Q3])*	13.8	[6.2–34.8]	19,4	[13.9–29.6]	*0.030*	32	[16.4–92.4]	*0.012*
* **Treatments** *								
Need for ICU admission *(n, %)*	29	(12.4%)	27	(42.9%)	*<0.0001*	9	(18.7%)	*0.007*
Antibiotic therapy *(n, %)*	217	(93.1%)	62	(98.4%)	*0.135*	45	(93.7%)	*0.314*
Corticosteroid therapy *(n, %)*	122	(52.4%)	41	(65.1%)	*0.077*	27	(56.2%)	*0.344*
Anticoagulant therapy *(n, %)*	219	(94.0%)	62	(98.4%)	*0.207*	45	(93.7%)	*0.314*
Length of stay, days *(median, [Q1–Q3])*	8	[5–14]	15	[10–32.5]	*<0.0001*	15.5	[10.2–29.0]	*0.847*

PRM = physical and rehabilitation medicine; PRU = polyvalent rehabilitation units; Q = quartile (Q1 = 25th percentile and Q3 = 75th percentile); GFR = glomerular filtration rate; CRP = C-reactive protein; ICU, intensive care unit. ^†^ 35 missing data, ^₮^ 39 missing data, ^¥^ 75 not applicable.

## Data Availability

In French law, the data is the property of the promoter and can’t be shared.

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
