# Peer review of "Characteristics of COVID-19 Inpatients in Rehabilitation Units during the First Pandemic Wave: A Cohort Study from a Large Hospital in Champagne Region"

_biology, 2022, doi:10.3390/biology11060937_

Round 1
Reviewer 1 Report
1- The title shall be revised and be more specific. Since there are several reports similar to the current manuscript, the authors should specify their title by expressing their study population ( University Hospital of Reims, north-eastern France).
2- The importance of such a report should be described in the introduction. What is the necessity of such a report in the University Hospital of Reims, north-eastern France, while there are several reports from other areas?
3- The period of time the researchers have studied the patients and the number of included patients are not satisfactory for the conclusion they have made. In similar reports, a wider period of time has been used. Is there any reason for that?
4- Several previous studies have not been discussed or even mentioned in the background and discussion. Some of those are PMID: 33558860,P MID: 33180754. The authors should improve their literature reviews.
5- The outcome and prognosis of the studied variables, either clinical or biological are to be analyzed in a cohort. I would like to strongly suggest improving the manuscript in this regard.
Author Response
REVIEWER 1
Review Report Form
Open Review
( ) I would not like to sign my review report
(x) I would like to sign my review report
English language and style
( ) Extensive editing of English language and style required
(x) Moderate English changes required
( ) English language and style are fine/minor spell check required
( ) I don't feel qualified to judge about the English language and style
|
Yes |
Can be improved |
Must be improved |
Not applicable |
|
|
Does the introduction provide sufficient background and include all relevant references? |
(x) |
( ) |
( ) |
( ) |
|
Are all the cited references relevant to the research? |
( ) |
(x) |
( ) |
( ) |
|
Is the research design appropriate? |
( ) |
(x) |
( ) |
( ) |
|
Are the methods adequately described? |
( ) |
(x) |
( ) |
( ) |
|
Are the results clearly presented? |
( ) |
(x) |
( ) |
( ) |
|
Are the conclusions supported by the results? |
( ) |
(x) |
( ) |
( ) |
Comments and Suggestions for Authors
1- The title shall be revised and be more specific. Since there are several reports similar to the current manuscript, the authors should specify their title by expressing their study population ( University Hospital of Reims, north-eastern France).
Response: We thank the Reviewer for the comments and useful suggestions, which have helped us to significantly improve the manuscript. We have changed the title to the following:
“Characteristics of COVID-19 inpatients in rehabilitation units during the first pandemic wave: A cohort study from a large hospital in Champagne region”
2- The importance of such a report should be described in the introduction. What is the necessity of such a report in the University Hospital of Reims, north-eastern France, while there are several reports from other areas?
Response: Again, thank you for raising this important question and suggestion. Referral to rehabilitation is a specific component of healthcare organization, with a specific population of patients. How best to select these patients is not always clear to the medical community. This is why we chose to study this specific population in our institution. The text has been modified as follows (line 91):
“In this context of emergency re-organization, and without knowing the outcome in the post-acute phase, the medical community did its utmost to treat patients as well as possible. Patients who were referred to rehabilitation were the patients who were deemed to have the highest priority. The study of these criteria would make it possible to understand the choices made, and to analyze the relevant factors driving those choices, as well as areas where there may be room for improvement in this context. Previous studies have investigated the characteristics of patients admitted to the ICU, who were discharged and readmitted to an acute care unit [18–20]. Wiertz et al. described the characteristics of patients admitted to rehabilitation, but without comparing them to patients not referred to rehabilitation [21].”
3- The period of time the researchers have studied the patients and the number of included patients are not satisfactory for the conclusion they have made. In similar reports, a wider period of time has been used. Is there any reason for that?
Response: The Reviewer raises a relevant point. The objective of the study was to investigate the characteristics associated with referral to PRM units at the time of the pandemic, and the acute adaptation of caregivers in this situation. The period selected corresponds to the most intense period of the first wave in France, with a subsequent drop in the number of patients admitted, due to the effects of the national lockdown that was implemented in France. Since the epidemic wave began earlier in the East of France, there was a marked decrease in patient influx to the University Hospital of Reims after the 30th April. We have added a few lines in the revised manuscript, line 108, to explain this choice, to take account of your comment, as follows:
“Due to the fact that the pandemic began first in France in the northeastern and north-central departments, with a temporal progression from these departments towards the rest of country [22], the period selected corresponds to the most intense period of the first wave, and there was a marked drop in the number of patients being admitted after the 30th April, as the effects of the national lockdown began to become apparent.”
4- Several previous studies have not been discussed or even mentioned in the background and discussion. Some of those are PMID: 33558860,P MID: 33180754. The authors should improve their literature reviews.
Response: Thank you for bringing these interesting publications to our notice. We have revised the literature cited in the introduction and discussion, and included a few lines about the papers you cited, as follows:
- One sentence has been added to the introduction, line 96 :
“Previous studies have investigated the characteristics of patients admitted to the ICU, who were discharged and readmitted to an acute care unit [18–20]. Wiertz et al. described the characteristics of patients admitted to rehabilitation, but without comparing them to patients not referred to rehabilitation [21].”
- We have also added some text to the discussion:
Line 233:
“Several studies have sought to quantify rehabilitation needs after COVID-19. Halpin et al. reported 60.2% of patients with fatigue and 42.6% with dyspnea between 4 and 8 weeks after discharge [28]. Nakayama and al. reported that 62.6% of patients had persistent symptoms after discharge [29]. Regarding participation, Fugazzaro et al found that 76% of patients had participation restrictions, as assessed by the Reintegration to Normal Living Index, at 3 months after the acute infection [30]. In light of these proportions, it seems insufficient that only one fifth of patients are receiving PRM. This shortfall was also highlighted by Daunter et al. who reported that 45.2% of patients had functional decline impacting their discharge after hospitalization, and 40.6% 40.6%) were never assessed by a PRM physician, physical therapist, occupational therapist, or speech therapist during their hospital stay [31].”
Line 250:
“Older patients, and patients at risk of severe forms of disease due to cumulative risk factors, were more often referred to PRM units in the present study, whereas patients with a single risk factor for disease severity, such as obesity or diabetes, were less frequently referred. A potential explanation can be found in the frailty definition, related to age and comorbidities. Hägg et al. [34] showed an association between the level of frailty, as measured by the Clinical Frailty Scale, and the discharge after hospitalisation for COVID-19 disease. Comorbidities were not associated with discharge to home in that study, which is consistent with our results.”
5- The outcome and prognosis of the studied variables, either clinical or biological are to be analyzed in a cohort. I would like to strongly suggest improving the manuscript in this regard.
Response: We do thank the Reviewer for this constructive proposal, and we did indeed give a great deal of thought to the usefulness of univariate analysis. However, although such a model might have predictive value, it would not be generalizable to other populations in view of the single-centre nature of our cohort. Therefore, we believe that the low likelihood of robust results that could be extrapolated to other contexts precludes the performance of statistical comparisons, as they would not contribute to the scientific community
Submission Date
16 May 2022
Date of this review
26 May 2022 11:16:16

Reviewer 2 Report
In this manuscript, several concerns need to be addressed as follows:
1. The introduction is very concise and has not covered the topic properly as follows:
- The authors should describe the medical procedures needed for the rehabilitation of COVID-19 patients.
- The description of The University Hospital of Reims (lines 63-67) should be transferred to the method section.
- The biological role of renal biomarker and troponin in COVID-19.
2. Material and methods:
- What about the exclusion criteria.
- The methods used for estimation of the hematological and biochemical indicators should be mentioned.
3. Lines 240-243 and 263-264: it seems like part of the template of the journal.
4. It is not preferred to begin sentences with abbreviations like PRM in line 41. Please revise the whole manuscript for such an error.
5. The manuscript needs to be revised for the English and the overall writing style. The writing style should be formal from the third-person perspective. Do not use we or our.
6. There is a problem with using abbreviations throughout the manuscript. The full term should be mentioned first with the abbreviation between paresis then the abbreviations should be exclusively used throughout the manuscript. E.g., in line 56, the intensive care unit has been abbreviated as ICU then the full term has been repeated again in lines 65 and 116. Such errors have been repeated for many abbreviations throughout the manuscript.
Author Response
REVIEWER 2
Review Report Form
Open Review
( ) I would not like to sign my review report
(x) I would like to sign my review report
English language and style
(x) Extensive editing of English language and style required
( ) Moderate English changes required
( ) English language and style are fine/minor spell check required
( ) I don't feel qualified to judge about the English language and style
|
Yes |
Can be improved |
Must be improved |
Not applicable |
|
|
Does the introduction provide sufficient background and include all relevant references? |
( ) |
( ) |
(x) |
( ) |
|
Are all the cited references relevant to the research? |
( ) |
( ) |
(x) |
( ) |
|
Is the research design appropriate? |
(x) |
( ) |
( ) |
( ) |
|
Are the methods adequately described? |
( ) |
( ) |
(x) |
( ) |
|
Are the results clearly presented? |
(x) |
( ) |
( ) |
( ) |
|
Are the conclusions supported by the results? |
(x) |
( ) |
( ) |
( ) |
Comments and Suggestions for Authors
In this manuscript, several concerns need to be addressed as follows:
- The introduction is very concise and has not covered the topic properly as follows:
Response: We would like to thank the Reviewer for the constructive comments and useful suggestions, which have helped us to improve the manuscript. Please find our point by point answers below regarding the introduction:
- The authors should describe the medical procedures needed for the rehabilitation of COVID-19 patients.
Response: Thank you for this useful suggestion. COVID-19 is responsible for multisystemic impairment, most frequently resulting in residual fatigue, muscular weakness, exercise intolerance, and cognitive decline. Consequently, rehabilitation must be individualized, and must be multicomponent in nature, to cater for each patient’s specific deficits. To take the Reviewer’s comment into account, we have added the following sentence (line 83):
“Therefore, PRM specialists expected to receive many patients within a short time, all requiring personalized and multicomponent rehabilitation, due to the wide heterogeneity in symptoms.”
- The description of The University Hospital of Reims (lines 63-67) should be transferred to the method section.
Response: Thank you for this suggestion. However, the description provided was intended to describe the organization of the French hospital system, with acute medicine-surgery-obstetrics units on the one hand, and post-acute care and rehabilitation units on the other hand. The organization of the University Hospital of Reims is consistent with the national organization. This clarification is made in the introduction in order to introduce the terms used in the article and to inform the reader about how the health system works in our country. We have rephrased the sentence to make this clearer (line 71).
“The organisation of the public hospital system in France provides The University Hospital of Reims in northeastern France has a first-line reserve of acute hospitalization wards in medicine, surgery and obstetrics, including medical and surgical ICUs. Second-line resources include as polyvalent rehabilitation units (PRU), or specialized rehabilitation units (e.g. Physical and Rehabilitation Medicine (PRM) units).”
- The biological role of renal biomarker and troponin in COVID-19.
Response: We agree with the Reviewer that renal biomarkers and troponin have a role, and therefore, we have included them in the introduction, line 58, as follows:
“Biological markers, such as troponin or renal function, are also related to severity of infection [4,5].”
- Material and methods:
- What about the exclusion criteria.
Response: We have clarified in the methods section that there were no specific exclusion criteria (line 129):
There were no specific exclusion criteria.
- The methods used for estimation of the hematological and biochemical indicators should be mentioned.
Response: Thank you for bringing this omission to our notice. We have added this information in the revised version (line 148), as follows:
Hematological indicators (leukocytes and lymphocytes) were estimated using the Sysmex®XN1000 analyzer (Sysmex, Roissy CDG, France). All biochemical indicators were estimated using the Roche analyzer with Roche reagents (Roche Diagnostics, Meylan, France). Creatinine was estimated with an enzymatic assay protocol, and glomerular filtration rate (GFR) was calculated using the CKD-EPI equation. C-reactive protein (CRP) and albuminemia were estimated by immunoturbidimetry. Troponin was estimated by electrochemiluminescence.
- Lines 240-243 and 263-264: it seems like part of the template of the journal.
Response: Thank you for your careful proofreading. These parts have been removed from the manuscript.
- 4. It is not preferred to begin sentences with abbreviations like PRM in line 41. Please revise the whole manuscript for such an error.
Response: Thank you again for the thorough proofreading. This error has been corrected, in line 41, and throughout the whole manuscript.
- The manuscript needs to be revised for the English and the overall writing style. The writing style should be formal from the third-person perspective. Do not use we or our.
Response: The manuscript has been thoroughly revised by a native English-speaking medical writer.
- There is a problem with using abbreviations throughout the manuscript. The full term should be mentioned first with the abbreviation between paresis then the abbreviations should be exclusively used throughout the manuscript. E.g., in line 56, the intensive care unit has been abbreviated as ICU then the full term has been repeated again in lines 65 and 116. Such errors have been repeated for many abbreviations throughout the manuscript.
Response: Thank you again for the thorough proofreading. We have carefully corrected the use of the abbreviations throughout the manuscript.
Submission Date
16 May 2022
Date of this review
24 May 2022 16:07:22

Reviewer 3 Report
STRUCTURE
- The manuscript is properly structured. However, it would be convenient to delete the "simple summary", incorporating the objectives of the study in the "introduction".
- Throughout the article, the authors' contributions are included in different sections:
o line 240: “Authors should discuss the results and how they can be interpreted from the perspective of previous studies and of the working hypotheses. The findings and their implications should be discussed in the broadest context possible. Future research directions may also be highlighted”.
o line 263: “This section is not mandatory but can be added to the manuscript if the discussion is unusually long or complex”.
Please review it and remove the respective comments.
TITLE AND ABSTRACT
- The Abstract is properly structured.
INTRODUCTION
- Most of the citations are more than 5 years old. The references should be updated.
- Perhaps it would be useful to begin the introduction with a brief description of COVID-19 and main characteristics.
- The literature search is brief. Are there other factors related to survival rate (type of cancer, cardiovascular risks, age, etc.)?
o Slim K, Boirie Y. The quintuple penalty of obese patients in the COVID-19 pandemic. Surg Obes Relat Dis. 2020 Aug;16(8):1163-1164. doi: 10.1016/j.soard.2020.04.032. Epub 2020 May 1. PMID: 32418769; PMCID: PMC7252000.
o Zhou F, Yu T, Du R, Fan G, Liu Y, Liu Z, Xiang J, Wang Y, Song B, Gu X, Guan L, Wei Y, Li H, Wu X, Xu J, Tu S, Zhang Y, Chen H, Cao B. Clinical course and risk factors for mortality of adult inpatients with COVID-19 in Wuhan, China: a retrospective cohort study. Lancet. 2020 Mar 28;395(10229):1054-1062. doi: 10.1016/S0140-6736(20)30566-3. Epub 2020 Mar 11. Erratum in: Lancet. 2020 Mar 28;395(10229):1038. Erratum in: Lancet. 2020 Mar 28;395(10229):1038. PMID: 32171076; PMCID: PMC7270627.
- It is recommended to present the situation of other hospital centers and the claims of some health professions.
o Park S, Elliott J, Berlin A, Hamer-Hunt J, Haines A. Strengthening the UK primary care response to covid-19. BMJ. 2020 Sep 25;370:m3691. doi: 10.1136/bmj.m3691. PMID: 32978177.
o Tran BX, Vo LH, Phan HT, Pham HQ, Vu GT, Le HT, Latkin CA, Ho CS, Ho RC. Mobilizing medical students for COVID-19 responses: Experience of Vietnam. J Glob Health. 2020 Dec;10(2):020319. doi: 10.7189/jogh.10.020319. PMID: 33110521; PMCID: PMC7559422.
MATERIAL AND METHODS
Study population
- Line: 92: The reporting is informed by the Strengthening the Reporting of Observational studies in Epidemiology (STROBE) statement for cohort studies”. The guide used should be referenced.
- How was the cohort followed up?
- What were the methods of data collection? were they validated? (blood tests, weight, height, etc.)?
Study measurements
- Line 110: Comorbidities (with calculation of Charlson’s comorbidity index). References this method.
- Line 123: The initial clinical presentation was considered to be severe in case of fever and : < 90% in room air, or breathing rate > 30/min, or need for invasive or non-invasive ventilation, or in case of respiratory or cardio-circulatory organ failure. Reference this statement to give more credibility to the study.
- : Mention the meaning of the acronyms the first time they are referred to in the text.
Statistical analysis
- Which statistical program has been used?
- Groups b and c are not compared
- Explain how the quartiles were divided.
- Specify the following sentence: Quantitative data are described as median and quartiles, and qualitative data as number (percentage).
RESULTS
- The first table should only provide data on the characteristics of the participants (demographics, age, gender, etc.). Are there statistically significant differences?
- Line 168: Table 1. When a table is split into two sheets, the sections must be put back in the header. Applicable to the rest of the manuscript.
- Blood parameter data are not discussed.
DISCUSSION
- Provide a cautious overall interpretation of the results taking into account the objectives, multiplicity of analyses, results of similar studies, and other relevant evidence.
CONCLUSION
- What kind of solutions can be applied (policies, programs, etc.)?
o World Health Organization. (2020). Strengthening the health system response to COVID-19: technical guidance# 1: maintaining the delivery of essential health care services while mobilizing the health workforce for the COVID-19 response, 18 April 2020 (No. WHO/EURO: 2020-669-40404-54161). World Health Organization. Regional Office for Europe.
REFERENCES
- References follow the indicated style
Author Response
REVIEWER 3
Review Report Form
Open Review
( ) I would not like to sign my review report
(x) I would like to sign my review report
English language and style
( ) Extensive editing of English language and style required
( ) Moderate English changes required
(x) English language and style are fine/minor spell check required
( ) I don't feel qualified to judge about the English language and style
|
Yes |
Can be improved |
Must be improved |
Not applicable |
|
|
Does the introduction provide sufficient background and include all relevant references? |
( ) |
(x) |
( ) |
( ) |
|
Are all the cited references relevant to the research? |
( ) |
(x) |
( ) |
( ) |
|
Is the research design appropriate? |
( ) |
(x) |
( ) |
( ) |
|
Are the methods adequately described? |
( ) |
(x) |
( ) |
( ) |
|
Are the results clearly presented? |
( ) |
(x) |
( ) |
( ) |
|
Are the conclusions supported by the results? |
( ) |
(x) |
( ) |
( ) |
Comments and Suggestions for Authors
STRUCTURE
- The manuscript is properly structured. However, it would be convenient to delete the "simple summary", incorporating the objectives of the study in the "introduction".
Response: We thank the Reviewer for the pertinent comments and useful suggestions, which have helped us to improve the manuscript.
Regarding the simple summary, the journal specifically requires inclusion of a 200-words simple summary in lay language. Therefore, we are unable to remove this part.
Nevertheless, we have stated the objectives more clearly in the introduction, as follows (line 101):
The aim of this study is to describe the profiles of COVID-19 patients referred to rehabilitation units during the first epidemic wave in the University Hospital of Reims, France, to better understand the rehabilitation needs during a pandemic crisis.
- Throughout the article, the authors' contributions are included in different sections:
o line 240: “Authors should discuss the results and how they can be interpreted from the perspective of previous studies and of the working hypotheses. The findings and their implications should be discussed in the broadest context possible. Future research directions may also be highlighted”.
o line 263: “This section is not mandatory but can be added to the manuscript if the discussion is unusually long or complex”.
Please review it and remove the respective comments.
Response: Thank you for the careful proofreading! We apologies for omitting to delete the parts, which were the instructions included in the manuscript template. These parts have now been removed.
TITLE AND ABSTRACT
- The Abstract is properly structured.
Response: Thank you for your positive appreciation.
INTRODUCTION
- Most of the citations are more than 5 years old. The references should be updated.
Response : The references have been revised, and we have included several new references in the introduction and discussion, in response to your suggestion and a similar remark from another Reviewer.
- Perhaps it would be useful to begin the introduction with a brief description of COVID-19 and main characteristics.
Response : Thank you for this useful suggestion. We have added the following text (line 54):
“COVID-19 is a respiratory infection caused by the SARS-CoV-2 virus, and the main symptoms are fever and cough. About 15% of patients with symptomatic infection will have a severe form of infection, and around 5% require intensive care [1].
- The literature search is brief. Are there other factors related to survival rate (type of cancer, cardiovascular risks, age, etc.)?
- Slim K, Boirie Y. The quintuple penalty of obese patients in the COVID-19 pandemic. Surg Obes Relat Dis. 2020 Aug;16(8):1163-1164. doi: 10.1016/j.soard.2020.04.032. Epub 2020 May 1. PMID: 32418769; PMCID: PMC7252000.
- Zhou F, Yu T, Du R, Fan G, Liu Y, Liu Z, Xiang J, Wang Y, Song B, Gu X, Guan L, Wei Y, Li H, Wu X, Xu J, Tu S, Zhang Y, Chen H, Cao B. Clinical course and risk factors for mortality of adult inpatients with COVID-19 in Wuhan, China: a retrospective cohort study. Lancet. 2020 Mar 28;395(10229):1054-1062. doi: 10.1016/S0140-6736(20)30566-3. Epub 2020 Mar 11. Erratum in: Lancet. 2020 Mar 28;395(10229):1038. Erratum in: Lancet. 2020 Mar 28;395(10229):1038. PMID: 32171076; PMCID: PMC7270627.
Response: Thank you for this relevant suggestion. Indeed, other factors have been identified in the literature, and these are now mentioned with relevant references (line56):
“Several factors have been shown to be related to mortality, such as obesity, arterial hypertension, diabetes, and cardiovascular disease [2,3]. Biological markers, such as troponin or renal function, are also related to severity of infection [4,5].
.
- It is recommended to present the situation of other hospital centers and the claims of some health professions.
o Park S, Elliott J, Berlin A, Hamer-Hunt J, Haines A. Strengthening the UK primary care response to covid-19. BMJ. 2020 Sep 25;370:m3691. doi: 10.1136/bmj.m3691. PMID: 32978177.
o Tran BX, Vo LH, Phan HT, Pham HQ, Vu GT, Le HT, Latkin CA, Ho CS, Ho RC. Mobilizing medical students for COVID-19 responses: Experience of Vietnam. J Glob Health. 2020 Dec;10(2):020319. doi: 10.7189/jogh.10.020319. PMID: 33110521; PMCID: PMC7559422.
Response: Thank you for this international perspective, which we now mention, line 65, as follows:
“Different countries have implemented various adaptation strategies to strengthen the capacity to respond to the epidemic, for example by mobilizing medical students, or through cooperation with primary care [8,9].”
MATERIAL AND METHODS
Study population
- Line: 92: The reporting is informed by the Strengthening the Reporting of Observational studies in Epidemiology (STROBE) statement for cohort studies”. The guide used should be referenced.
Response: The reference to the STROBE statement has been added.
von Elm, E.; Altman, D.G.; Egger, M.; Pocock, S.J.; Gøtzsche, P.C.; Vandenbroucke, J.P.; STROBE Initiative The Strengthening the Reporting of Observational Studies in Epidemiology (STROBE) Statement: Guidelines for Reporting Observational Studies. Int J Surg 2014, 12, 1495–1499, doi:10.1016/j.ijsu.2014.07.013.
- How was the cohort followed up?
Response: We thank the Reviewer for raising this important point. All patients were identified via the hospital informatics database, and baseline information was collected retrospectively. According to the protocol registered with ClinicalTrials.gov (under the identifier NCT04553575), patients from the cohort were followed-up at 6, 12 and 24 months after diagnosis, with a medical consultation at each follow-up timepoint.
The details of the follow-up have been added to the text, line 132, as follows:
“According to the protocol, patients were followed-up at 6, 12 and 24 months after diagnosis, with a medical consultation at each follow-up timepoint.”
- What were the methods of data collection? were they validated? (blood tests, weight, height, etc.)?
Response: Data were collected retrospectively from the patients’ computerized medical records. The biological data in the medical files were all validated by the physicians responsible for the laboratory at the time of analysis, before being entered into the medical file. Anthropometric data, such as weight and height, were re-read and verified by the caregivers working with the patient.
The manuscript has been modified to make this clearer, line 137, as follows:
“Data were retrospectively collected from the patients’ computerized medical records. The biological data in the medical files were all validated by the physicians responsible for the laboratory at the time of analysis, before being entered into the medical file. Anthropometric data, such as weight and height, were re-read and verified by the caregivers working with the patient.”
Study measurements
- Line 110: Comorbidities (with calculation of Charlson’s comorbidity index). References this method.
Response: The following reference has been added:
Tuty Kuswardhani, R.A.; Henrina, J.; Pranata, R.; Anthonius Lim, M.; Lawrensia, S.; Suastika, K. Charlson Comorbidity Index and a Composite of Poor Outcomes in COVID-19 Patients: A Systematic Review and Meta-Analysis. Diabetes Metab Syndr 2020, 14, 2103–2109, doi:10.1016/j.dsx.2020.10.022.
- Line 123: The initial clinical presentation was considered to be severe in case of fever and : < 90% in room air, or breathing rate > 30/min, or need for invasive or non-invasive ventilation, or in case of respiratory or cardio-circulatory organ failure.
Response: We thank the Reviewer for this important point. The criteria used are the severity criteria proposed at the beginning of the first wave by the French Society of Anesthesia & Intensive Care Medicine (SFAR). The reference has been added and the following sentence has been added to the text (line 160):
“In accordance with the recommendations of the French Society of Anesthesia & Intensive Care Medicine (SFAR) [26], initial clinical presentation was considered to be severe in case of:”
- : Mention the meaning of the acronyms the first time they are referred to in the text.
Response: Thank you for the careful proofreading. The whole manuscript has been proofread and all abbreviations and acronyms have been defined at first use.
Statistical analysis
- Which statistical program has been used?
Response: Data were analysed with R Studio®. This has been specified in the methods section as follows (line 179):
All analyses were conducted using R studio® Version 4.0.5.
- Groups b and c are not compared
Response: We thank the Reviewer for pointing this out. We compared groups B and C (in the last column of the table). In fact, we did not include a comparison between groups A (patients discharged to home) and groups B+C (not discharged to home) since our objective was centered on patients admitted to PRM units. Nevertheless, these results are provided in the supplementary material, and this is mentioned Line 209:
“A comparison of patients discharged to home and those referred to PRU + PRM is shown in the supplemental material.”
- Explain how the quartiles were divided.
Response: The descriptive statistics used are described in the text (line 173) and in the legend of the tables, as follows:
Quantitative data are described as median and first and third quartiles (25th and 75th percentiles).
- Specify the following sentence: Quantitative data are described as median and quartiles, and qualitative data as number (percentage).
Response: This sentence has been rephrased, in line with the previous comment. The distribution of the data was tested for normality using the Shapiro-Wilk test. In case of a non-normal distribution, the data were described as median with the first and third quartiles (Q1 = 25th percentile, Q3 = 75th percentile).
RESULTS
- The first table should only provide data on the characteristics of the participants (demographics, age, gender, etc.). Are there statistically significant differences?
Response: The original Table 1 has now been divided into 2 tables. The information regarding the significant results has been specified in the text as follows (line 194):
Patients addressed to PRM units were older and more at risk of severe form of disease than patient discharged to home. Patients addressed to PRU were older and had a higher BMI than patients referred to PRM units.
- Line 168: Table 1. When a table is split into two sheets, the sections must be put back in the header. Applicable to the rest of the manuscript.
Response: Thank you for this suggestion, we have applied it throughout the text.
- Blood parameter data are not discussed.
Response : thank you for pointing out this omission. We have added the following sentence in the results, line 201, to give the blood parameters:
Patients with acute renal failure were more frequently referred to PRM units, and had a higher level of troponin at the acute phase than patients discharged to home, but had lower CRP and troponin levels than patients referred to PRU.
DISCUSSION
- Provide a cautious overall interpretation of the results taking into account the objectives, multiplicity of analyses, results of similar studies, and other relevant evidence.
Response: Thank you for pertinent advice. The discussion has been revised, as follows:
Line 233:
“Several studies have sought to quantify rehabilitation needs after COVID-19. Halpin et al. reported 60.2% of patients with fatigue and 42.6% with dyspnea between 4 and 8 weeks after discharge [28]. Nakayama and al. reported that 62.6% of patients had persistent symptoms after discharge [29]. Regarding participation, Fugazzaro et al found that 76% of patients had participation restrictions, as assessed by the Reintegration to Normal Living Index, at 3 months after the acute infection [30]. In light of these proportions, it seems insufficient that only one fifth of patients are receiving PRM. This shortfall was also highlighted by Daunter et al. who reported that 45.2% of patients had functional decline impacting their discharge after hospitalization, and 40.6% 40.6%) were never assessed by a PRM physician, physical therapist, occupational therapist, or speech therapist during their hospital stay [31].”
Line 246:
“The association of ICU and PRM admission is consistent with rehabilitation needs after ICU [32]. The longer median length of stay results in prolonged immobility with the potential consequences such as sarcopenia [33], creating further rehabilitation needs.
Line 250:
“Older patients, and patients at risk of severe forms of disease due to cumulative risk factors, were more often referred to PRM units in the present study, whereas patients with a single risk factor for disease severity, such as obesity or diabetes, were less frequently referred. A potential explanation can be found in the frailty definition, related to age and comorbidities. Hägg et al. [34] showed an association between the level of frailty, as measured by the Clinical Frailty Scale, and the discharge after hospitalisation for COVID-19 disease. Comorbidities were not associated with discharge to home in that study, which is consistent with our results.”
Line 259:
“Different mechanisms may explain the increase in troponin level at the acute phase of COVID-19, mainly non-ischaemic myocardial processed due to the severity of infection [35]. Acute renal failure is also related to severity, and age, which is consistent with the higher age in PRM patients [36]. These results support the association between disease severity and referral to PRM units. However, other biomarkers associated with disease severity, such as albumin or inflammatory markers [3], were not associated with rehabilitation. Thus, severity may not be the only explanatory factor.
Line 279:
The problem of the availability of places during the first wave has also been studied in other types of unit, such as ICU, with a need for rationing [6].
Line 283:
“Ahmad et al. [39] described a post-COVID care center that was established to identify patients with disease sequalae with a view to delivering early multidisciplinary rehabilitation services. Since not all rehabilitation needs can be covered, systematic evaluation or open consultations should be developed within an overall strategy to offer early rehabilitation interventions [40], and to maintain the continuum of care [41].”
CONCLUSION
- What kind of solutions can be applied (policies, programs, etc.)?
o World Health Organization. (2020). Strengthening the health system response to COVID-19: technical guidance# 1: maintaining the delivery of essential health care services while mobilizing the health workforce for the COVID-19 response, 18 April 2020 (No. WHO/EURO: 2020-669-40404-54161). World Health Organization. Regional Office for Europe.
Response: Thank you for opening up this perspective about the potential solutions that can be applied, as we transition to a chronic phase of COVID-19.
Propositions to improve practices include the need for further studies investigating acute variables related to functional impairment; the development of early consultations for patient not immediately referred to PRM units; and systematic functional evaluation at the acute phase before discharge.
This reference has been used line 288.
The conclusion has been rephrased to add these points, as follows (line 328):
Organisational conditions for healthcare delivery and availability of services/beds, or the patient’s functional capacity must be a concern. Pandemic strategies for adapting access to rehabilitation should be developed, such as early consultation and multidisciplinary evaluation of discharged patients, to maintain the continuum of care.
REFERENCES
- References follow the indicated style
Response: Thank you.
Submission Date
16 May 2022
Date of this review
30 May 2022 09:48:43

Round 2
Reviewer 1 Report
The authors addressed all comments and answered all questions.
Reviewer 2 Report
No further comments to be addressed
Reviewer 3 Report
No further comments.